# Intercalation events visualized in single microcrystals of graphite

Edward R. White[1], Jared J. Lodico[1] & B.C. Regan[1]

The electrochemical intercalation of layered materials, particularly graphite, is fundamental to the operation of rechargeable energy-storage devices such as the lithium-ion battery and the carbon-enhanced lead-acid battery. Intercalation is thought to proceed in discrete stages, where each stage represents a specific structure and stoichiometry of the intercalant relative to the host. However, the three-dimensional structures of the stages between unintercalated and fully intercalated are not known, and the dynamics of the transitions between stages are not understood. Using optical and scanning transmission electron microscopy, we video the intercalation of single microcrystals of graphite in concentrated sulfuric acid. Here we find that intercalation charge transfer proceeds through highly variable current pulses that, although directly associated with structural changes, do not match the expectations of the classical theories. Evidently random nanoscopic defects dominate the dynamics of intercalation.

[1] Department of Physics & Astronomy and California NanoSystems Institute, University of California, Los Angeles, CA 90095, USA. Edward R. White and Jared J. Lodico contributed equally to this work. Correspondence and requests for materials should be addressed to E.R.W. (email: ewhite@physics.ucla.edu) or to B.C.R. (email: regan@physics.ucla.edu)

Graphite and other layered materials (e.g., transition metal dichalcogenides) can reversibly absorb and disgorge large amounts of charge[1, 2]. In the forward process, known as intercalation, ions move between and separate the van der Waals-bonded layers that constitute the host crystal, while leaving the individual layers themselves relatively unchanged. In some applications, the resulting intercalation compounds (ICs) serve as precursors for producing exfoliated layered materials[3]. In other applications (e.g., energy storage) the reaction is reversed. Upon deintercalation the interlayer galleries empty and the host reverts approximately to its initial structure. The properties of reversibility and large charge capacity make graphite ICs and other related materials well-suited for application as electrodes in rechargeable batteries (both as anodes and as cathodes[2, 4]) and supercapacitors[5].

Although electrochemical intercalation has been known since the 19th century and is fundamental to the operation of, for example, lithium-ion and carbon-enhanced lead-acid batteries, the structure of ICs is not well understood. X-ray and electron diffraction data indicate that in many ICs[6] the intercalants organize in layers, and that these intercalant layers are themselves regularly spaced[7]. Transitioning from pristine to fully intercalated, a host crystal passes through stages, where 'stage $n$' indicates $n$ host layers separating each intercalant layer[1, 7]. Thus, stage 1 is fully intercalated and stage 2 is half-intercalated. First proposed by Rüdorff and Hofmann (RH), the simplest structural model for staged ICs features intercalant layers that span the entire crystal host. However, this naive model gives a physically unreasonable picture of any $\Delta n = 1$ stage transition other than the $1 \leftrightarrow 2$ transition, as was noted by the proposers themselves[7].

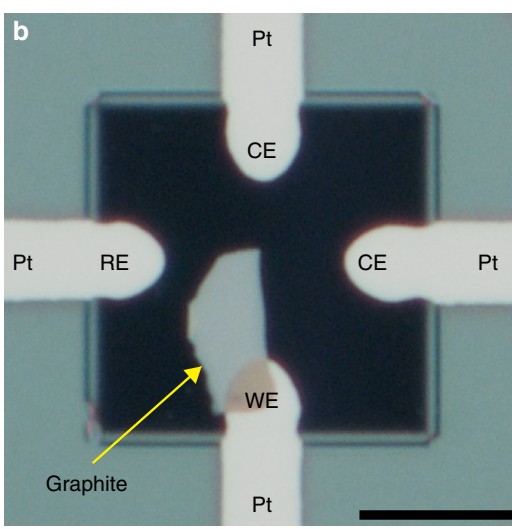

**Fig. 1** Fluid cell construction. **a** Exploded schematic of the electrochemical fluid cell. A drop of $H_2SO_4$ (disk) is sandwiched between two electron-transparent $Si_3N_4$ windows framed by silicon chips. **b** Optical micrograph showing a graphite single crystal on a bottom-chip window, before the $H_2SO_4$ and the top chip are added. Four platinum contacts converge over the window; the working (WE), pseudo-reference (RE), and counter (CE) electrodes are labeled. The scale bar is 10 µm

To allow for the full variety of observed stage number transitions, Daumas and Hérold (DH) postulated[8] that for stage $n > 1$ intercalant layers do not span the crystal, but rather organize within a gallery into 'islands' that are small compared to the crystal area, and that these islands stack to form 'domains' of a given stage number[9–11]. According to DH theory, these domains reconfigure to affect a stage transition[12] by sliding or diffusing. The intercalant packing density within a domain is not necessarily constant, and in some circumstances is altered by an external control variable. For instance, the stage number can be changed while the IC stoichiometry is held constant[13], and vice versa[14], by varying the pressure[13] or electrical potential[14].

Although DH theory provides a reasonable alternative to the RH model, the structures that it predicts—the domains themselves[9] and the transition states[15, 16]—have proven difficult to observe. Evidence for the existence of DH islands (though, notably, not for regular staged domains) has been acquired with the cross-sectional transmission electron microscopy (TEM) of $FeCl_3$ intercalated into comminuted natural graphite[17]. But high-resolution TEM of commensurate $SbCl_5$ graphite ICs found more support for the RH model than the DH model[9]. Estimates for the size of the domains vary from molecular-scale[16] to >1 µm[3, 9–11]. Reconciling DH theory with the successful exfoliation of stage 3 and stage 2 graphite ICs to tri- and bi-layer graphene[3] is also problematic, as the atomic mechanism by which domains in the IC generate flakes of the dispersed product is not clear. Neither the RH model nor the DH model can be considered well-established[16].

In this communication, we examine a prediction that both models share: as a function of the electrochemical potential, the electrochemical currents associated with stage transitions $n \leftrightarrow n \pm 1$ are larger for $n$ smaller[14]. For instance, the stage sequence $3 \rightarrow 2 \rightarrow 1$ is expected to be associated with stored charge changes $\Delta q$ in the ratios $1/12:1/6:1/2$ relative to the full capacity of the electrode. For a sufficiently small and ideal host crystallite, these changes should be abrupt. Allowing for changes in the equilibrium intercalant packing density gives transitions that are less abrupt and $\Delta q$ ratios that are slightly modified, but the main qualitative feature remains: larger currents accompany lower stage number transitions[14].

Using optical microscopy and scanning transmission electron microscopy (STEM), we image small, high-quality single crystals of natural graphite in concentrated sulfuric acid undergoing electrochemical intercalation and deintercalation via cyclic voltammetry (CV). (The five movies included in the Supporting Information provide a summary of the results.) Our video data show contrast changes associated with electrochemical charge transfer that broadly reproduce, cycle after cycle. The details of the charge transfer, on the other hand, do not reproduce well from cycle to cycle, with the observed current pulses occurring in no evident pattern. In the STEM experiments we also observe irreversible contrast changes, most notable during the sample's first intercalation cycle, that are attributable to the intercalation processes and not to beam damage. Frequently, the reversible contrast changes span the sample, are abrupt, and are associated with identifiable electrochemical current pulses. Counter to the expectations of the DH and RH models, we do not see current pulses that are systematically larger for low-stage transitions. Even with high-quality microcrystals, graphite's intercalation dynamics seem to be dominated by kinetic, extrinsic factors such as defects, and not the intrinsic thermodynamics[11].

## Results

**Experimental setup.** For the in situ STEM experiments we fabricated sealed fluid cells[18–21] containing sulfuric acid (18 M $H_2SO_4$,

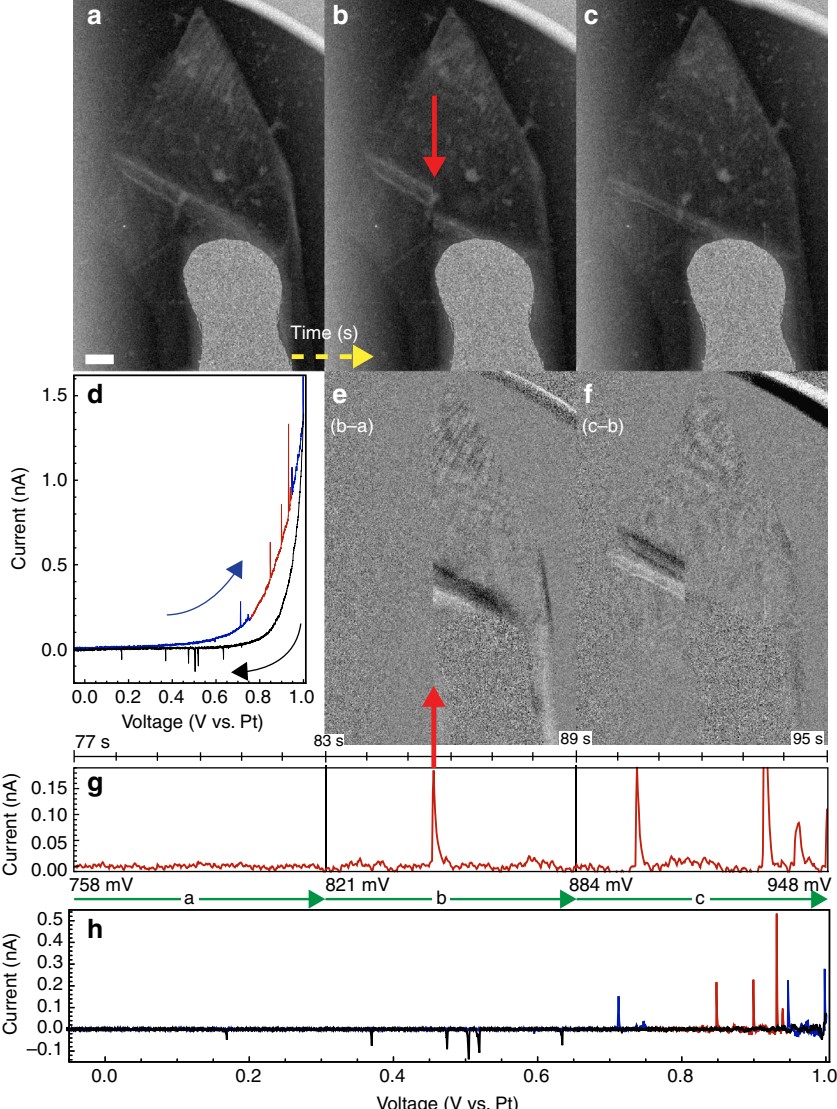

**Fig. 2** Intercalation events. **a**–**c** Three sequential ADF STEM images of a single crystal graphite flake in a fluid cell. (Supplementary Movie 2 is a more complete description of this experiment.) Here the images are oriented such that the beam raster proceeded bottom-to-top and then left-to-right. Structural changes in the graphite flake occurred during the acquisition of **b** at the time indicated by the red arrow. **d** Cyclic voltammogram obtained during this experiment. Arrows indicate the ramp direction of the voltage during intercalation (blue) and deintercalation (black). **e**, **f** Computed difference images: (**b**–**a** and **c**–**b**) highlight the changes between these sequential frames. **g** Voltage applied (constant ramp rate of $\pm10\,\mathrm{mV\,s^{-1}}$) and current (background-subtracted) measured during the acquisition of **a**–**c**, with the abscissa scaled to match the timing of **a**–**c** and **e**, **f**. An intercalation current peak (containing 28 pC, equivalent to 1.5 crystal-spanning intercalant layers) is coincident with the structural changes that occur during the acquisition of **b**. **h** The voltammogram for the complete cycle **d**, after background current subtraction. The scale bar is 500 nm

96%), pre-patterned platinum electrodes and single-crystal graphite flakes (typical crystallite size $\sim 100\,\mu\mathrm{m}^2 \times 10\,\mathrm{nm}$) produced via mechanical exfoliation from bulk natural graphite (Naturgraphit GmbH). Figure 1 shows a schematic of the STEM cell architecture and a typical device. (The optical microscopy experiments used a similar architecture but were unsealed.) The crystallinity of the flakes was verified with TEM before the cells were assembled (Supplementary Fig. 1). The cells had platinum working (WE), pseudo-reference, and counter (CE) electrodes. (We performed control experiments with bulk graphite that indicate that platinum is a stable reference 14 for this system—see Supplementary Fig. 2 and Supplementary Movie 1. More information is available in the Methods section and Supporting Information.).

This system turns out to be nearly ideal for in situ STEM studies; imaging for hours over many intercalation/deintercalation cycles produced only minor beam-induced sample damage

and contamination. This chemistry is also well-suited for the problem at hand: the intercalation of graphite with sulfuric acid is known to be easy and orderly[11], and is of direct relevance to the performance of hybrid supercapacitors and carbon-enhanced lead-acid batteries[5, 22]. Li-ion batteries should show much the same intercalation physics[1], although the lithium/graphite system has a cationic intercalant and an aprotic electrolyte, forms a solid electrolyte interface layer, exhibits dendrite growth, and is air-sensitive, to name a few important differences[23]. However, it is hoped that the simpler, more straightforward model system studied here represents an easier route toward both a general understanding of intercalation, and the development of powerful in situ TEM techniques that can be applied to the problem.

**Intercalation events**. Figure 2a shows an annular dark field (ADF) STEM image of a $68\,\mu\mathrm{m}^2 \times 8\,\mathrm{nm}$ graphite flake inside a

fluid cell. The flake is electrically connected to the platinum lead extending from the bottom of the image, and contacts sulfuric acid outside the field of view. Its CV (Fig. 2d) shows the current gradually increasing (decreasing) as the WE potential is ramped up (down). Superimposed on the gradual changes are sharp peaks. The gradual current variation is seen in control experiments lacking a graphite flake (Supplementary Fig. 3), and is attributed to double layer charging of the platinum electrode. To isolate the graphite electrochemistry we subtract this smoothly varying background current (see Supplementary Fig. 4). As we explain below, the sharp peaks that remain (Fig. 2h) can be attributed to intercalation events in the graphite.

Figure 2a–c show three STEM images, sequentially acquired during the CV of Fig. 2d. STEM images are conventionally oriented such that the first pixel acquired appears in the upper left corner and the last acquired appears in the lower right. Here the images are rotated counter-clockwise 90° relative to this convention; the electron beam rastered bottom-to-top, and then left-to-right. Displayed in this way, each column of pixels moving left-to-right across Fig. 2a–c was acquired 512 pixels × 30 μs pixel$^{-1}$ = 15.4 ms after its predecessor. The electrical current data (background-subtracted) collected during the acquisition of the Fig. 2a–c STEM images (Fig. 2g) is aligned and scaled such that the current and the image data share the same (horizontal) time axis. In this way changes in the ADF image that occur during acquisition can be directly correlated with the current peaks in the CV.

Part way through the acquisition of Fig. 2b the graphite changes suddenly relative to Fig. 2a; the feature extending from the tip of the platinum contact to the left edge of the graphite (perhaps a wrinkle) shifts closer to the bottom of the image, and the narrow bright and dark bands near the tip of the graphite spread to show more uniform intensity. These changes are highlighted in Fig. 2e, the difference image calculated by subtracting Fig. 2a from b. Bright (dark) pixels thus correspond to an increase (decrease) in ADF intensity in the second frame (Fig. 2b) relative to the first (Fig. 2a). The simultaneously

acquired current data show a sharp peak that is coincident with the onset of the Fig. 2b event, indicating an association with the change in the graphite.

The next image, Fig. 2c, reveals that the remainder of the wrinkle feature moved as well. The corresponding difference image (Fig. 2f) shows changes in the feature until the pixel column associated with the event in Fig. 2b. Since the latter portion of Fig. 2f shows little contrast variation, the structural changes in the wrinkle feature were evidently abrupt compared to the 5.9 s frame time.

During the acquisition of Fig. 2c several more current peaks occur. The first may be associated with the further spreading of the bright and dark bands near the tip of the graphite. The remaining events occur when the STEM raster has nearly reached the right edge of the graphite, and is thus blind to changes over most of the crystallite (until the next frame, when the raster re-starts). The subsequent frame (see Supplementary Movie 2) shows the complete removal of the wrinkle feature, likely a result of the events near the end of Fig. 2c.

Optical microscopy of graphite crystallites undergoing CV in sulfuric acid shows similar correlations between current peaks and graphite contrast changes. Figure 3 summarizes experiments on a flake with 16 times the area and about 10 times the thickness of the one shown in Fig. 2. Experiments at this intermediate scale (Fig. 3 flake mass 0.2 ng) help substantiate the extrapolation of lessons learned in the STEM experiments (e.g., Figure 2 flake mass 1.2 pg) to bulk applications. The CV (Fig. 3d) shows current spikes on top of gradual current changes, much like Fig. 2d. Optical micrographs of the graphite flake acquired immediately before (Fig. 3a) and after (Fig. 3b) a current spike (red in Fig. 3d, e) show a color shift that is easily seen in the computed difference image (Fig. 3c). This color shift can be attributed to intercalation; graphite, gray in its pristine state, becomes blue when intercalated with sulfuric acid[7, 16].

The abrupt contrast changes evident in both the STEM images (Fig. 2) and the optical images (Fig. 3) are evidently associated with structural changes in the graphite crystallites. The correlated

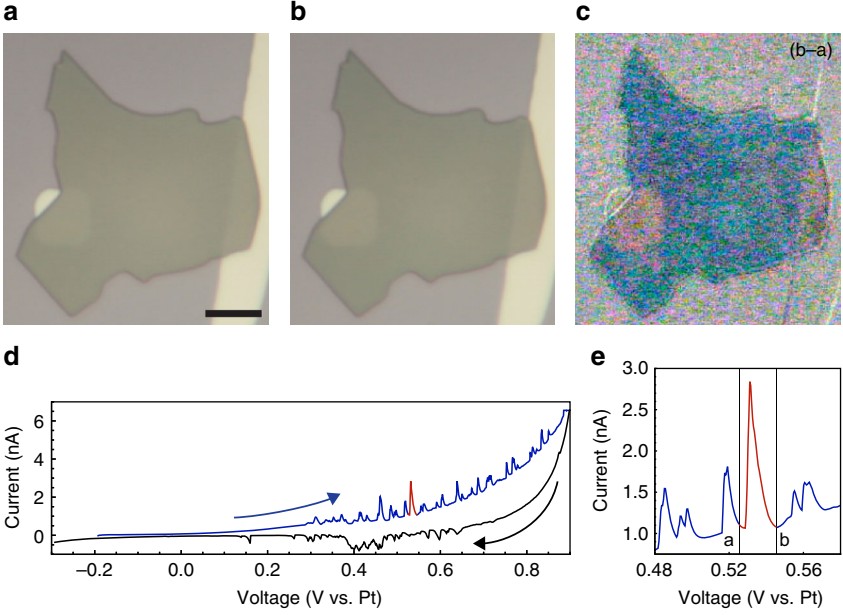

**Fig. 3** Optical microscopy of intercalation event. **a**, **b** Two sequential photographs of a single crystal graphite flake as it undergoes an intercalation event. **c** Computed difference image (b−a) showing the graphite flake's color change. **d** Full cyclic voltammogram for the complete intercalation/deintercalation cycle. Highlighted in red is the current peak associated with the intercalation event, which contains 2.0 nC, equivalent to 4.3 intercalant layers. **e** Section of the full cycle in **d** emphasizing the current peak. The black bars indicate when photographs (**a**, **b**) were acquired. The scale bar is 10 μm

current spikes and contrast/structural changes we term intercalation events. Their presence in both optical and STEM experiments indicates that they are fundamental, and exist independent of any electron beam- or light-induced effects. We attribute their visibility here to the use of very small, high-quality, single-crystal graphite flakes. (In some cases we see current spikes without obvious structural changes, and in others we see structural changes with no clearly associated current spikes. Thus, the correlations are not perfect, perhaps in part because of experimental limitations such as the slow image acquisition rate and electronic noise in the electrochemical circuit.) In larger, more defective, and more polycrystalline samples, bulk averaging would necessarily obscure these events in the CV data. Moreover, observing the structural changes requires video microscopy, which is not a standard accompaniment to CV measurements. Thus it is unsurprising that such intercalation events have not been reported previously.

**Irreversible structural changes in folded graphite**. During the first intercalation cycle, some intercalation events are associated with irreversible structural changes. Figure 4 shows a graphite flake that is folded onto itself, creating moiré fringes in the STEM images. In the unintercalated graphite (Fig. 4a) the undulating appearance of the moiré fringes was likely the result of wrinkling in the graphite flake[24]. During the first CV, the moiré fringe irregularity decreased (Fig. 4b) before any structural changes elsewhere in the flake (see Supplementary Movie 3 and Supplementary Fig. 5a).

We imaged several flakes with folds, and in every case the first structural change occurred in the overlapping region (see Supplementary Fig. 5). For a random fold angle the top and bottom sheets generally have imperfect registry at the fold interface, and a concomitant decrease in the van der Waals binding between the interface layers. A correspondingly smaller energy barrier to entry into the gallery at the fold interface would explain the observed lower intercalation potential. Once intercalants have entered the gallery, defects between the sheets are unpinned and the fold can achieve better registry (Fig. 4b). Generalizing this observation, we conjecture that when multiple graphite flakes are in close contact, as in, for example, a battery electrode[25], intercalation between the flakes occurs before the bulk intercalation of the constituent crystals. Thus, an electrode's 'bulk' intercalation potential could be engineered to smaller values by re-stacking single layers with layer-to-layer misorientations.

Further electrochemical manipulation of the flake removes more defects. After completing the first full intercalation/ deintercalation cycle the moiré fringes are almost perfectly straight (Fig. 4c). Several circular defects evident near the center of Fig. 4a, b have also disappeared. Structures similar to these circular defects have previously been identified as dislocation loops[26], though making such an identification here leaves open the question of how intercalation removes the implied vacancies or interstitials (though the disappearances of the loops in sets of two could indicate pair annihilation). The elimination of these defects and irregularities is irreversible; subsequent, repeated cycling does not return the moiré pattern to its defective initial state (see Supplementary Movie 3).

**Defect re-organization over multiple cycles**. The graphite flakes continue to evolve structurally with repeated cycling. Figure 5 shows the same flake as pictured in Fig. 4, but away from the fold. The pristine single crystal graphite flake (Fig. 5a) contains many defects, as indicated by the dark lines within the crystal[24]. With

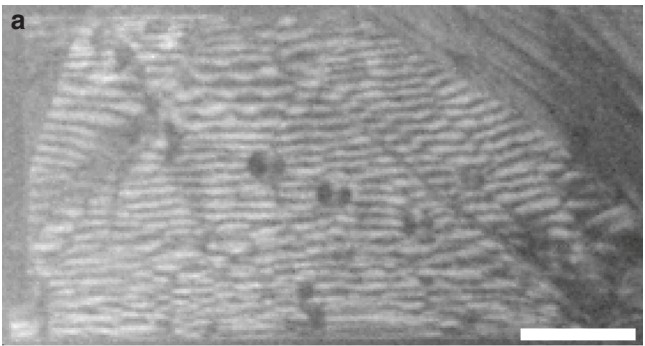

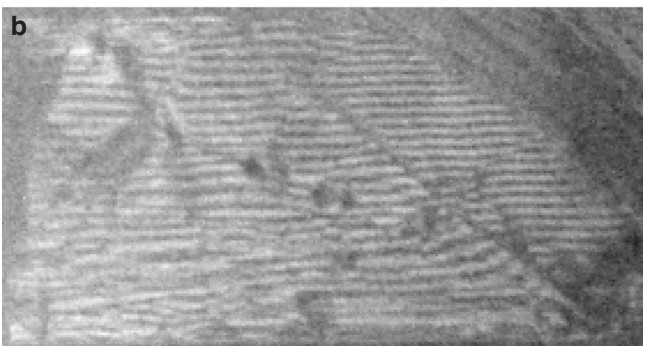

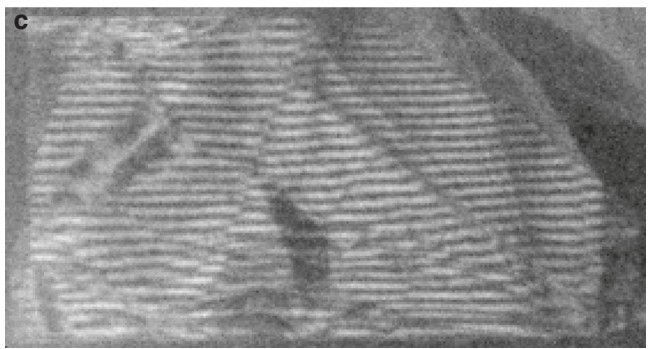

**Fig. 4** Moiré pattern evolution during intercalation. Moiré pattern from a 10 nm-thick graphite flake folded onto itself: (**a**) pristine, (**b**) after the first structural change and (**c**) after one complete intercalation/deintercalation cycle. See Supplementary Movie 3 for the complete evolution. The scale bar is 200 nm

intercalation cycling the defects reconfigure, move, and in some cases disappear (Fig. 5b–f).

Compared with many chemically-active systems, the graphite/ $H_2SO_4$ system is quite immune to beam-induced effects. Intercalation-induced and beam-induced structural changes are easily distinguished in these experiments: the former generally occur abruptly (and reversibly) over a large area of the crystallite, whereas the latter arise gradually and are most noticeable as tiny deposits that only grow over the course of an experiment. Although we cannot rule out the possibility that the electron beam precipitates abrupt events (for example, the focused probe may release defects pinning graphite layers), we note that in such a case the charge transfer would be dominated by defects even in the absence of the triggering electron beam, with larger thresholds. (Such a difference in the transport between the beam-on and beam-off conditions was, however, not observed—see Supplementary Fig. 6.) After STEM imaging the sample of Figs 4 and 5 for >2 h, the intercalation-induced contrast changes were still large, whereas the beam-induced deposition and damage were almost unnoticeable (see Supplementary Movie 3). The evident stability of this system with respect to imaging at 300 kV is

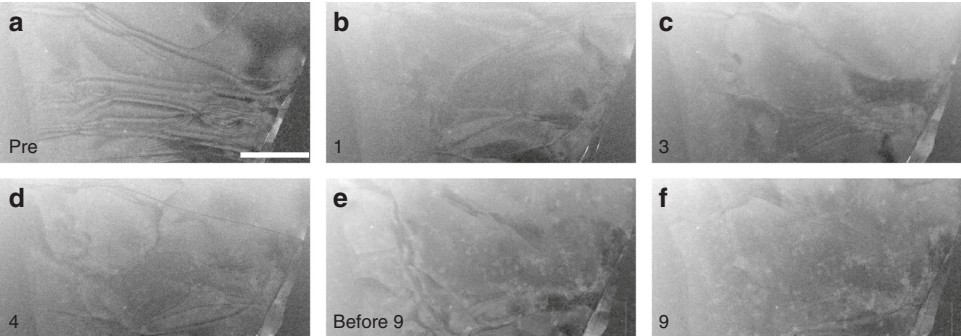

**Fig. 5** Structural changes through many intercalation/deintercalation cycles. **a** A pristine graphite flake prior to any intercalation. **b–f** The same graphite flake at the end of complete intercalation/deintercalation cycles 1, 3, and 4, at the beginning of cycle 9, and at the end of cycle 9, as indicated in the bottom left corner. During cycles 5–8 other areas of this flake were imaged at high magnification. See Supplementary Movie 3. The scale bar is 500 nm

surprising, given the knock-on damage that occurs in graphene above 80 kV ref. 27. We attribute this welcome result to the use of STEM with small beam currents (~ 30 pA) at relatively low magnification[28], the use of many-layer flakes, and the acidic environment[29].

The low beam currents also mitigate any beam-induced effects in the CV measurements. These effects, when they occur, are easily recognized, as they add a periodicity to the measured current that is synchronous with the STEM frame acquisition time. In other words, the beam's contribution to the current measured in the electrochemical circuit varies with the beam position. To make identifying such beam-induced currents straightforward, we are careful to always include graphite and graphite-free (i.e., silicon nitride membrane only) regions in the STEM field of view.

**Changing ADF STEM signal during intercalation.** Changes in the STEM contrast, often in the form of regular bright and dark bands, usually accompany current pulses evident in the transport data. Figure 6 documents several such intercalation events in a graphite microcrystal. According to estimates based on the flake size and the integrated charge transfer, the intercalation reaction proceeded here from the unintercalated state (Fig. 6a) to a mixture of stage 4 and stage 3 (Fig. 6c).

When unintercalated (a), the graphite had relatively uniform ADF intensity. Intercalation event I, which occurred as the acquisition of image (a) was completing, produced the broad bright and dark bands in the graphite evident in the subsequent frame (b). Intercalation events II and III caused minor changes to the bands, and intercalation event IV made the bright and dark bands smaller (c). The characteristic length scale for a bright–dark oscillation is more than one micrometer in (b), whereas it is only a few hundred nanometers in (c). This progression toward smaller bright–dark oscillations with decreasing stage numbers is consistently evident in our STEM intercalation experiments (see Supplementary Movies 2, 4, and 5).

The bright–dark contrast variation is not yet understood, but these plan-view images might be indicating the presence of DH domains. The bands of Fig. 6b, c appear to be bend contours[30]. The graphite flake cannot be bending uniformly, because uniform bending would alter the thickness of the fluid cell and produce a change in the background signal that is not observed: the insets in Fig. 6b, c show no contrast change away from the graphite. However, bends within the graphite of alternating direction are expected on opposing sides of intercalant domains[1, 8]. At the edges of a domain the two graphite layers must bend back together around the intercalant as the interlayer spacing changes from 7.98 Å to 3.345 Å ref. 7. Such bends might be responsible for the observed bright-dark contrast variation. Observations of the

bands disappearing upon reaching stage 1 would support this interpretation, but unfortunately such complete intercalation has proven difficult to achieve in our sealed *in situ* STEM fluid cells, perhaps because of gas evolution. In any case, if correct, this interpretation indicates that the characteristic lateral size of the domains is decreasing with decreasing stage number $n \geq 2$, a trend not predicted by the DH model. Note that the large aspect ratio of the bands is not necessarily indicative of the same in the domains themselves; if the electron beam is not at perfect normal incidence upon the graphite, some bending directions will produce more contrast variation than others.

**Intercalation current peaks and theoretical predictions.** The RH and DH models make quantitative predictions for the electrochemical currents associated with each stage transition. According to the simplest versions of these models, each stage transition is coincident with a single electrochemical pulse[14] (Fig. 7a). The charge expected in each current peak is $\frac{Q}{n(n+1)}$, where $Q$ is the full capacity of the electrode, and $n$ and $n + 1$ refer to the two stages involved in the transition $n \leftrightarrow n + 1$. The intercalation and deintercalation current peaks are equal in number and size, and decreasing $n$ yields successively larger peaks.

The intercalation of bulk graphite (Fig. 7b) gives data meeting these general expectations: the stage $2 \rightarrow 1$ and $1 \rightarrow 2$ transitions have larger current peaks than any other stage change, and the peak sizes are similar[14]. However, single graphite microcrystals do not exhibit the expected intercalation behavior. As shown in Fig. 7c, although the background-subtracted current peaks account for the majority of the total intercalation charge transfer, the peaks individually are generally not large enough to cause stage transitions. Instead, low-stage transitions are usually composed of many current peaks, which is to say that many intercalation or deintercalation events are required to cause a stage transition. On the other hand, a few events are associated with transitions across multiple stages. Intercalation events corresponding to a single-stage transition are rare. Moreover, the current peak size does not always increase with decreasing stage number, and the size distribution changes from cycle to cycle. Finally, the expected symmetry between intercalation and deintercalation is not present: the deintercalation events greatly outnumber the intercalation events. In summary, we observe current pulses that, in aggregate, account for the graphite's intercalation and deintercalation. However, these pulses do not show the size distribution, nor even the symmetry between the forward and reverse processes, that are expected according to the classical intercalation models. Control experiments with the electron beam blanked show the same behavior (Supplementary Fig. 6).

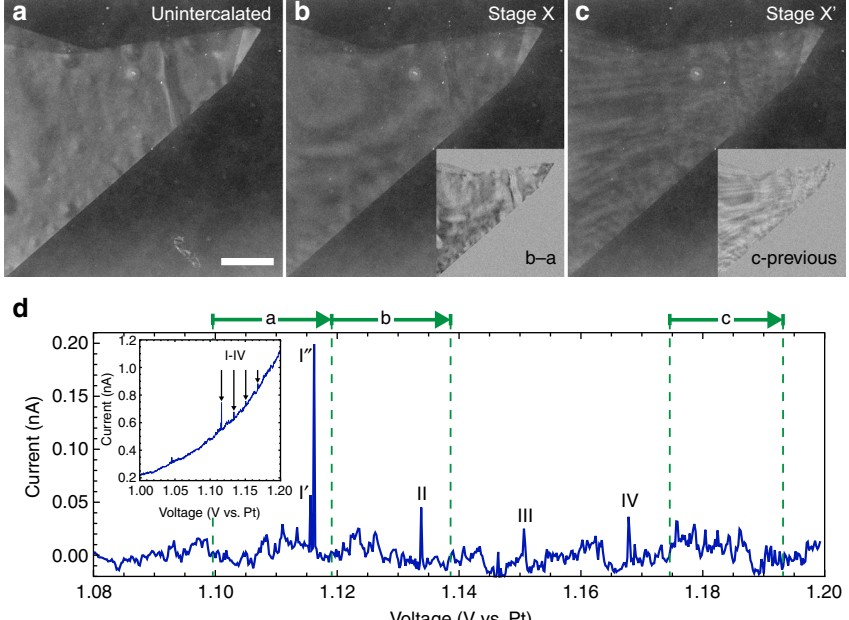

**Fig. 6** STEM contrast evolution with decreasing stage number. **a–c** ADF STEM images of unintercalated (**a**), stage X (**b**), and stage X′ < X (**c**) graphite. The insets are calculated difference images between the displayed and its preceding frame. **d** Background-subtracted current vs. voltage. Individual intercalation events are labeled I–IV. Events I′ and I″, referred to collectively as event I in the text, occur 60 ms apart (equivalent to 15 STEM pixel rows). The voltage range over which images **a–c** are acquired is indicated by the green arrows. The inset shows the current-voltage data before subtraction. (For a discussion of the small peak just below 1.05 V see Supplementary Fig. 5b.) See Supplementary Movie 4. The scale bar is 1 μm

Thus, the RH and DH intercalation models fail to describe the transport in a single microcrystal, an extremely simple, nearly ideal system where these classical models are expected to be most successful. Although bulk data (Fig. 7b) show qualitative agreement with the model predictions (Fig. 7a), the single microcrystal data (Fig. 7c) does not. We attribute the discrepancy to random defects playing a surprisingly dominant role.

## Discussion

These defects govern transport at the crystallite level, but their effects average out over the many microcrystals in a bulk electrode. Highly oriented pyrolytic graphite is known to be more defective and more difficult to intercalate than single crystal graphite[1, 2, 10], in agreement with this hypothesis. A defect-containing region that would otherwise intercalate to stage $n$ at voltage $V$, might, because of an additional energy barrier, only reach stage $n$ at voltage $V + \Delta V$. Thus, in a region with multiple defects a stage $n$ transition would be completed gradually though many intercalation events as the voltage surpassed the thresholds corresponding to the individual defects. If a defect or series of defects presented a barrier large enough, a single intercalation event could bring about a multiple stage transition ($\Delta n \geq 2$).

Such a defect-dominated charge transfer model can qualitatively explain the data of Fig. 7c, which shows many small current pulses corresponding to fractional transitions, and a few large pulses corresponding to transitions across multiple stages. Intercalation current pulses are generally larger than deintercalation pulses, implying that the defect energy barriers for intercalation are higher than for deintercalation. The irreproducibility of the current peaks indicates that the defects are not forming in exactly the same way from cycle to cycle. Such irreproducibility was also observed in our optical experiments, indicating that the important defect production mechanism is intrinsic to the graphite–sulfuric acid system and not the result of electron beam irradiation. Variability is also evident in the STEM imaging data: six images of a graphite flake (Fig. 5) acquired at the same

(unintercalated) point in an intercalation/deintercalation cycle each show a unique configuration of defects. Thus both transport and imaging data support a model of charge transfer in microcrystals that is dominated by random defects.

In conclusion, using video optical and electron microscopy we have identified electrochemical events that occur during the intercalation and deintercalation of single-microcrystal graphite in concentrated sulfuric acid. These intercalation events consist of electrical current pulses associated with abrupt structural changes in the microcrystal, and in sum account for most of the complete chemical reaction that moves the microcrystal back and forth between the intercalated and unintercalated states. The first intercalation events occur where a microcrystal flake has folded back upon itself, a priority attributable to the imperfect crystalline registry at the interface between the two folds. The particular defects within a microcrystal, as revealed by STEM imaging, vary from cycle to cycle, as do the electrochemical current pulses. Within a given cycle the arrangement of current pulses is markedly asymmetric between intercalation and deintercalation, which also indicates that non-thermodynamic, extrinsic factors are controlling. Thus even in high-quality single crystals, intercalation transport seems dominated by defects; the expectations of the classical RH and DH models are met only with bulk averaging. Finally, the STEM contrast evolution with intercalation, while not yet understood, might be revealing the dynamics of domain organization in plan view, in which case it suggests that lateral domain size decreases with decreasing stage number $n \geq 2$.

## Methods

All STEM images were acquired with a Fischione Model 3000 ADF detector in a FEI Titan 80–300 kV TEM operated at an accelerating voltage of 300 kV with a ~ 30 pA beam current (spot 8, 50 μm C2 aperture). The camera length and convergence angle (2.9 mrad) were chosen to maximize the ratio of the ADF detector signal from the first-order relative to the second order graphite Bragg peaks, while excluding the zeroth order peak entirely. Although the small convergence angle precludes high-resolution imaging, these imaging conditions give both enhanced

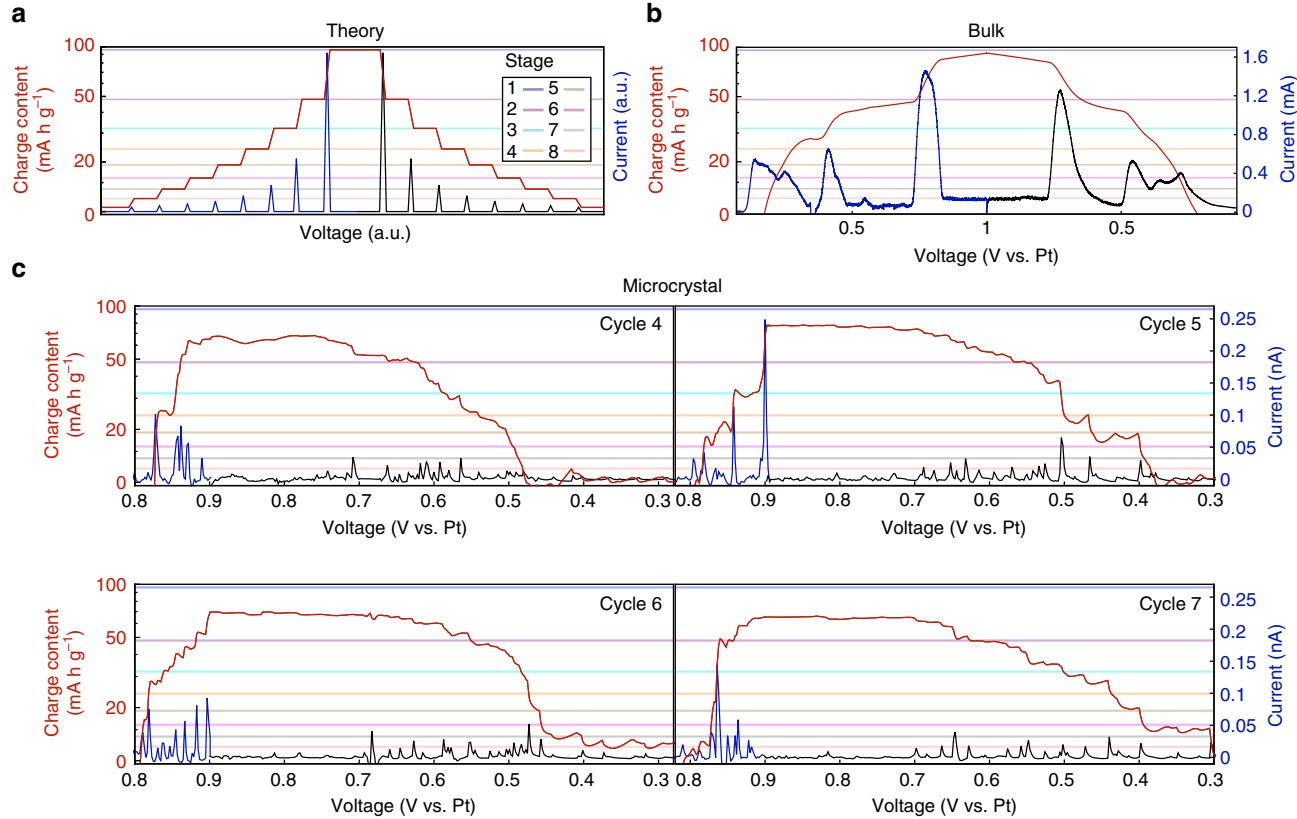

**Fig. 7** Current and charge content vs. voltage for the intercalation/deintercalation of graphite. The blue (black) curves represent positive (negative) current. The red curves give the charge content, calculated by integrating the current. Current is plotted on a linear scale and charge content on a logarithmic scale (to better separate high stage numbers). The specific capacities corresponding to stages 1–8 are indicated by the colored lines. **a** Model intercalation/deintercalation cycle[14] consistent with the RH and DH theories. **b** Intercalation/deintercalation cycle of bulk graphite (see Supplementary Fig. 2). **c** Four consecutive intercalation/deintercalation cycles from a single graphite microcrystal. The background current has been subtracted (for the raw CVs and sequential cycles see Supplementary Figs 7 and 8, respectively)

sensitivity to AB-to-AA stacking changes[31] (thought to sometimes occur in the stage 1 to stage 2 transition[1]) and diffraction contrast (see Supplementary Fig. 9), which were more valuable for this study. Samples were imaged at approximately normal incidence with respect to the electron beam, except for those shown in Supplementary Movies 3 and 5, Figs 4 and 5, and Supplementary Fig. 5a, which were tilted to maximize the signal in one first-order peak in the convergent beam electron diffraction pattern. This tilt probably explains why the moiré pattern at the fold in Fig. 4 and Supplementary Fig. 5a appears as straight lines[32] instead of the hexagonal pattern expected from an untilted sample (Supplementary Fig. 5c).

For the in situ STEM experiments the fluid cells were electrically contacted using a TEM biasing holder manufactured by Hummingbird Scientific. A Gamry 600 potentiostat was used to perform CV. A buffered output signal from the potentiostat was digitized in parallel with the signal from the STEM detector, allowing pixel-by-pixel temporal correlation of the electrical transport and the STEM image data.

**Data availability**. Data supporting the findings of this study are given in the paper and its Supplementary Information files. Raw data can be obtained from the corresponding authors.

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

## Acknowledgements

We thank Matthew Mecklenburg, Guillaume Muller and Bruce Dunn for valuable comments and discussions, and William Hubbard for assistance with chip fabrication. This work was supported in part by FAME, one of six centers of STARnet, a Semiconductor Research Corporation program sponsored by MARCO and DARPA, by National Science Foundation (NSF) award DMR-1611036, and by NSF STC award DMR-1548924. We acknowledge the use of instruments at the Electron Imaging Center for NanoMachines supported by NIH 1S10RR23057 and the CNSI at UCLA.

## Author contributions

E.R.W. and J.J.L. performed the experiments and analyzed the data. The manuscript was written with contributions from all authors.

## Additional information

**Competing interests:** The authors declare no competing financial interests.

**Change history:** The Peer Review File associated with this Article was updated shortly after publication to redact confidential comments to the editor.

