## [Peer Review File · Nature Communications]

Reviewers' comments:

Reviewer #1 (Remarks to the Author):

This paper reports the results obtained for the in-situ observations of the intercalation/deintercalation phenomena occurred in graphite microcrystals by plan-view imaging with the scanning transmission electron microscopy and the optical microscopy. The authors show the contrast changes in the images and the sharp current pulses in the CV curves by utilizing very thin, high-quality single crystals of graphite as the specimens. By such local observations and measurements (not in bulk specimens), they insisted to find the new intercalation phenomena controlled by non-thermodynamic factors, which cannot be explained by the classical theories. Based on these results, the authors predicted a new mechanism of the intercalation, that is dominantly controlled by random defects within the crystal. This conclusion itself is interesting, but some of in-situ data seems to be ambiguous, and thus the origin/structure of the random defects is not clear. The referee also has comments as followings.

- 1) The STEM images represent some characteristic contrasts such as bright/dark bands, paired circles and dark lines (in Fig. 2, 4, 5, 6). The authors insist that these contrasts arise from the defects within the graphite crystal according to the previous reports (e.g. ref. 23, 25). In these studies, however, the graphite defects had been observed with the conventional bright-field TEM. The (low-angle) dark-field STEM imaging used here must provide different image contrast. Since the role of the defects is one of the most important issue here, the authors should consider these points quantitatively.
- 2) The authors mention that the electron beam has almost no effects on the intercalation events observed in the present experiments. However, there is possibility that due to the electron beam, for example, the motion of the intercalants may be accelerated (or decelerated) and the barrier to release pinning sites between the graphite sheets may be reduced (or rise), resulting in differences in the image contrasts caused by the intercalation events. Are variations of the image contrast observed during the in-situ experiments also detected in ex-situ experiments? (i.e. by comparison of the images taken between before and after intercalation events with no e-beam irradiation during the electrochemical reaction) Are there differences in the CV curves measured in between the beam-on and -off conditions?
- 3) The authors suggest that the bright and dark contrast in the STEM images, e.g. Fig. 6, may represent the domain structures. Does this mean that the bright (or dark) bands correspond to the region where intercalants exist, while the dark (or bright) ones correspond to the edges of the above region (or non-intercalated region)? Please explain the reason of the contrast in detail. But, if so, the aspect ratio of each domain is relatively large, for example, 10 or more in Fig. 6c, though the graphite does not have such distorted structure. This argument is ambiguous.

Reviewer #2 (Remarks to the Author):

This article has creatively developed an in-situ STEM method with high time resolution to track

the structural changes in the graphite flakes during intercalation process, which is highly corresponding to the CV test. This method can determine the time nodes with high precision corresponding to the structural changes and the current pulses. This in situ experiment can potentially link the structural changes with the current pulses and explain the microscopic mechanism for the electrochemical performance. However, the spatial resolution is not high enough to directly resolve the detailed structural changes and the defects, which impose restrictions on the interpret ability. Furthermore, the interval time for the STEM image acquisition is relatively long and this will inevitably miss some important information of structural transformation. Some detailed concerns are listed below.

The details about the beam current are not explicitly explained.

1. The electron beam will also contribute some current during in situ STEM observation and CV tests. How to subtract this contribution?
2. Is the control CV experiment in Figure S3 beam-blanked or not blanked? Why is it so smooth if it is not blanked? If it is totally blanked, then it cannot be taken as a good control experiment, because it has not taken the beam current into consideration.
3. Why the current in the control CV experiment is four orders of magnitude higher than the optical and STEM experiment?
4. How to rule out the possibility that some of the current pulses are caused by electron beam and not due to the structural changes? Since the extra current generated by electron beam during STEM image acquisition could be as high as the current peaks observed in this article sometimes, this is a very important factor that need further consideration.

There are some doubts about the reversibility of the graphite flakes.

1. Is the intercalation process reversible in the graphite flakes? And how to determine the reversibility in this article? There seems to be two contradictory experimental phenomena: “contrast changes associated with stage transitions that broadly reproduces, cycle and cycle” and “the current peak size ... distribution changes from cycle to cycle”. If the contrast changes are associated with intercalation events, then the former supports the intercalation has good reversibility; if the current peaks are associated with intercalation events, then the latter phenomenon supports the intercalation has bad reversibility.
2. Furthermore, if both the current peaks and contrast changes are corresponding to the structural changes, why are they not corresponding to each other sometimes?

There are some doubts about the spatial resolution in determining the stage transition. 1. Why does this “intercalation events corresponding to a single stage transition are rare” happen? Could we deduce a conclusion (from this observation) that the intercalation in the microcrystal graphite flakes doesn't obey the stage transition routine as in the graphite bulk and there is no stage

transition herein? 2. Although this article focuses on the stage transition and its corresponding theory prediction, there is no direct and strong proof to prove that this microcrystal graphite flakes (with defects) are undergoing stage transitions during intercalation process. This article has linked the bright-dark contrast variation with the bends in the flakes caused by intercalation, however, the bright-dark contrast are not always formed as aligned strips as expected from the stage transition model.

Reviewer #3 (Remarks to the Author):

Graphite intercalation is studied with optical and transmission electron microscopy methods to examine the structural evolution of graphite in sulfuric acid. A succinct review of RH and DH theories of intercalation is presented, and sets the stage for the experiment study which aims to see if the island to domain transformation occurs, and then study the dynamics and critical points of such a transformation: the authors state that existing experimental studies indicate that the RH theory is more plausible than the DH. An intriguing introduction, the author's claim that, akin to dislocations dominating mechanical behavior in metals, defects dominate the intercalation behavior of even one of the most "perfect" experimental samples able to be tested. Whereas dislocations make metals "softer", defects make intercalates "slower"

Figure 2 is quite effective in conveying this information, if a bit dense. Combined with the evolution shown in Figure 4's and 5, the authors make a compelling case for future studies focussing on the defect structure of graphite in this system. While it is out of scope to show HOPG studies in this work, it compels the completion of that study to test the given hypothesis. The only sticky question left is whether or not the defects "iron themselves out" after many cycles. That is: after the initial cycling seen in this work, and the reconfiguration noted in the end of the results section, do the defects still play a role, or do they "get out of the way", say, at cycle 10 and beyond. The decrease in charge transfer resistance in real cells after many cycles might correlate with this.

Comments: > This chemistry is also well-suited for the problem at hand: the intercalation of graphite with sulfuric acid is known to be easy and orderly (11), and is of direct relevance to the performance of hybrid supercapacitors and carbon-enhanced lead-acid batteries (5,22)

A sentence or two comparing and contrasting the intercalation of H₂SO₄ in H₂O to Li from aprotic systems would be a key piece of information for readers coming from the Li Ion battery community.

Reviewers' comments:

Reviewer #1 (Remarks to the Author):

This paper reports the results obtained for the in-situ observations of the intercalation/deintercalation phenomena occurred in graphite microcrystals by plan-view imaging with the scanning transmission electron microscopy and the optical microscopy. The authors show the contrast changes in the images and the sharp current pulses in the CV curves by utilizing very thin, high-quality single crystals of graphite as the specimens. By such local observations and measurements (not in bulk specimens), they insisted to find the new intercalation phenomena controlled by non-thermodynamic factors, which cannot be explained by the classical theories. Based on these results, the authors predicted a new mechanism of the intercalation, that is dominantly controlled by random defects within the crystal. This conclusion itself is interesting, but some of in-situ data seems to be ambiguous, and thus the origin/structure of the random defects is not clear. The referee also has comments as followings.

1) The STEM images represent some characteristic contrasts such as bright/dark bands, paired circles and dark lines (in Fig. 2, 4, 5, 6). The authors insist that these contrasts arise from the defects within the graphite crystal according to the previous reports (e.g. ref. 23, 25). In these studies, however, the graphite defects had been observed with the conventional bright-field TEM. The (low-angle) dark-field STEM imaging used here must provide different image contrast. Since the role of the defects is one of the most important issue here, the authors should consider these points quantitatively.

We agree with the reviewer that a quantitative understanding of the image contrast is very desirable, and would help illuminate the structure of the defects. We have devoted some months of effort to obtaining a quantitative understanding, running thousands of multislice simulations using Christoph Koch's QSTEM package. Through this effort we have come to appreciate the difficulty of solving the “inverse problem” in the graphite-sulphuric acid system. In short, it is straightforward to simulate the image if the structure is known, but discovering the structure when you only have the image is a much more difficult problem. At present there are too many experimental uncertainties to make this problem tractable. While achieving a more quantitative understanding is an exciting prospect for future research, it is beyond the scope of the current paper.

2) The authors mention that the electron beam has almost no effects on the intercalation events observed in the present experiments. However, there is possibility that due to the electron beam, for example, the motion of the intercalants may be accelerated (or decelerated) and the barrier to release pinning sites between the graphite sheets may be reduced (or rise), resulting in differences in the image contrasts caused by the intercalation events. Are variations of the image contrast observed during the in-situ experiments also detected in ex-situ experiments? (i.e. by comparison of the images taken between before and after intercalation events with no e- beam irradiation during the electrochemical reaction)

The referee raises a valid point, and one that we attempted to address with Figure 3 (which

shows an intercalation event observed optically). We have added a sentence to the paragraph defining intercalation events to make this connection clearer.

As for the STEM data taken alone: we define an intercalation event as a current spike that is correlated with a contrast/structural change. We cannot measure the contrast or see a structural change without imaging the sample. Thus it is not possible for us to determine how the electron beam affects the intercalation events as observed by STEM alone.

Are there differences in the CV curves measured in between the beam-on and -off conditions?

Usually no, and when the beam does have an effect on the CV it is easily recognized. We have added a paragraph (“since they add a periodicity to the measured current that is synchronous with the STEM frame acquisition time.”) to explain. We have also added an additional figure (Supplementary Fig. 8) to the supplementary information along with a sentence to the discussion of Figure 7 (“control experiments”) to clarify.

3) The authors suggest that the bright and dark contrast in the STEM images, e.g. Fig. 6, may represent the domain structures. Does this mean that the bright (or dark) bands correspond to the region where intercalants exist, while the dark (or bright) ones correspond to the edges of the above region (or non-intercalated region)? Please explain the reason of the contrast in detail. But, if so, the aspect ratio of each domain is relatively large, for example, 10 or more in Fig. 6c, though the graphite does not have such distorted structure. This argument is ambiguous.

We have revised our discussion of Figure 6 to clarify our limited understanding of the contrast mechanism. In the process we added an explanation of the observed aspect ratios. A detailed, general explanation of the contrast mechanism is beyond our capabilities at present.

Reviewer #2 (Remarks to the Author):

This article has creatively developed an in-situ STEM method with high time resolution to track the structural changes in the graphite flakes during intercalation process, which is highly corresponding to the CV test. This method can determine the time nodes with high precision corresponding to the structural changes and the current pulses. This in situ experiment can potentially link the structural changes with the current pulses and explain the microscopic mechanism for the electrochemical performance. However, the spatial resolution is not high enough to directly resolve the detailed structural changes and the defects, which impose restrictions on the interpret ability. Furthermore, the interval time for the STEM image acquisition is relatively long and this will inevitably miss some important information of structural transformation. Some detailed concerns are listed below.

The details about the beam current are not explicitly explained.

1. The electron beam will also contribute some current during in situ STEM observation and CV tests. How to subtract this contribution?

See our penultimate response to Referee #1.

2. Is the control CV experiment in Figure S3 beam-blanked or not blanked? Why is it so smooth

if it is not blanked? If it is totally blanked, then it cannot be taken as a good control experiment, because it has not taken the beam current into consideration.

The control experiment in Supplementary Fig. 3 is done with no electron beam, and we have clarified this in the caption. As mentioned above, each STEM experiment has a built-in control; usually (though not always) we see no difference when the electron beam is on the graphite or far from it.

3. Why the current in the control CV experiment is four orders of magnitude higher than the optical and STEM experiment?

We thank the reviewer for noticing this discrepancy. That particular control experiment was done on a substrate that had not been sealed, and that had an inappropriately large area of the working electrode in contact with the electrolyte. We did the experiment using a more appropriately-sized drop of sulfuric acid and replaced the old figure with one showing the resulting data. The shape of the curves is gratifyingly similar.

4. How to rule out the possibility that some of the current pulses are caused by electron beam and not due to the structural changes? Since the extra current generated by electron beam during STEM image acquisition could be as high as the current peaks observed in this article sometimes, this is a very important factor that need further consideration.

Again, see our penultimate response to Referee #1.

There are some doubts about the reversibility of the graphite flakes.

1. Is the intercalation process reversible in the graphite flakes? And how to determine the reversibility in this article? There seems to be two contradictory experimental phenomena: “contrast changes associated with stage transitions that broadly reproduces, cycle and cycle” and “the current peak size ... distribution changes from cycle to cycle”. If the contrast changes are associated with intercalation events, then the former supports the intercalation has good reversibility; if the current peaks are associated with intercalation events, then the latter phenomenon supports the intercalation has bad reversibility.

We make a distinction between “broadly reproducible” and “reversible”. The process is reversible only insofar as the graphite can intercalate and deintercalate for many cycles. However, the details of the contrast changes and the current peak distribution change every cycle. We have attempted to clarify this point in the manuscript by revising the paragraph that begins, “Using optical microscopy and scanning transmission electron microscopy (STEM)...”

2. Furthermore, if both the current peaks and contrast changes are corresponding to the structural changes, why are they not corresponding to each other sometimes?

As the reviewer notes above, the STEM image acquisition rate is slow. Contrast changes that do not correspond to current peaks may be due to current pulses below our noise level. The former point is discussed in the paragraph describing Figure 2c. We clarify and emphasize both these points in new parenthetical remarks added to the paragraph in which we define intercalation

events.

There are some doubts about the spatial resolution in determining the stage transition. 1. Why does this “intercalation events corresponding to a single stage transition are rare” happen? Could we deduce a conclusion (from this observation) that the intercalation in the microcrystal graphite flakes doesn't obey the stage transition routine as in the graphite bulk and there is no stage transition herein?

Yes, that is exactly the main point of the paper.

2. Although this article focuses on the stage transition and its corresponding theory prediction, there is no direct and strong proof to prove that this microcrystal graphite flakes (with defects) are undergoing stage transitions during intercalation process. This article has linked the bright-dark contrast variation with the bends in the flakes caused by intercalation, however, the bright-dark contrast are not always formed as aligned strips as expected from the stage transition model.

We agree.

Reviewer #3 (Remarks to the Author):

Graphite intercalation is studied with optical and transmission electron microscopy methods to examine the structural evolution of graphite in sulfuric acid. A succinct review of RH and DH theories of intercalation is presented, and sets the stage for the experiment study which aims to see if the island to domain transformation occurs, and then study the dynamics and critical points of such a transformation: the authors state that existing experimental studies indicate that the RH theory is more plausible than the DH. An intriguing introduction, the author's claim that, akin to dislocations dominating mechanical behavior in metals, defects dominate the intercalation behavior of even one of the most “perfect” experimental samples able to be tested. Whereas dislocations make metals “softer”, defects make intercalates “slower”

Figure 2 is quite effective in conveying this information, if a bit dense. Combined with the evolution shown in Figure 4's and 5, the authors make a compelling case for future studies focussing on the defect structure of graphite in this system. While it is out of scope to show HOPG studies in this work, it compels the completion of that study to test the given hypothesis.

We agree, and thank the reviewer. It would be exciting to apply these methods to the study of HOPG.

The only sticky question left is whether or not the defects “iron themselves out” after many cycles. That is: after the initial cycling seen in this work, and the reconfiguration noted in the end of the results section, do the defects still play a role, or do they “get out of the way”, say, at cycle 10 and beyond. The decrease in charge transfer resistance in real cells after many cycles might correlate with this.

We agree. This an interesting question, and we will have to consider new experimental designs and procedures that will allow us to answer it within the constraints of performing in-situ TEM experiments.

Comments:

> This chemistry is also well-suited for the problem at hand: the intercalation of graphite with sulfuric acid is known to be easy and orderly (11), and is of direct relevance to the performance of hybrid supercapacitors and carbon-enhanced lead-acid batteries (5,22)

A sentence or two comparing and contrasting the intercalation of H₂SO₄ in H₂O to Li from aprotic systems would be a key piece of information for readers coming from the Li Ion battery community.

We thank the reviewer for this suggestion. To the paragraph beginning with “This system turns out to be nearly ideal” we have added two sentences (with a reference) describing the relevance of these experiments to the Li- graphite system.

Reviewers' comments (last round of peer review)

Reviewers' comments:

Reviewer #1 (Remarks to the Author):

The authors have prepared additional explanations and have revised their manuscript, according to my queries and requests. However, I have to report that it is difficult for me to accept the authors' answers and revisions because of the following reasons;

1) On the interpretation of ADF-STEM image contrasts related to defects in graphite; The authors state difficulties of image simulation for understanding the STEM image contrasts because of many experimental uncertainties. However, if it is not proved that the significant contrasts, i.e. paired circles and bright/dark bands, are coming from defects at least qualitatively, it is impossible to discuss about the relationship between the intercalation events and the behaviors of defects the authors have claimed. Based on such a speculation that the characteristic STEM contrasts are caused by the particular defects, the discussions and conclusions here should be also considered to be nothing more than speculations. I believe that a clear evidence is necessary on this point because the role of the defects is one of the most important issues in this article.

2) On the effects of electron beam irradiation; I agree with the authors that the intercalation events are fundamental because the structural changes and the correlated current spikes are detected in both STEM and OM observations. However, this does not lead to that e-beam does not affect to the reaction. The authors mentioned that beam-induced phenomena can be distinguished from intercalation-induced ones because beam-induced damages (e.g. changes to amorphous) arise gradually. I cannot neglect possibilities that abrupt events happen due to the e-beam, for example, focused electron probe may release pinning sites between the graphite sheets, which is not periodic phenomenon. Therefore, for evaluating the effect of e-beam, it is important to compare images taken before/after the intercalation reaction at regions where e-beam is irradiated / not irradiated during reaction.

3) On the domain structure appeared as bright/dark narrow bands; The authors explained that these contrasts in Fig.6 showing large aspect ratio may be produced by imperfect illumination tilted from normal incidence. However, this explanation is not valid because it's for the bright-field TEM imaging. The authors should discuss the contrast in the ADF-STEM imaging mode they have utilized.

Reviewer #2 (Remarks to the Author):

The authors have addressed my previous concerns; thus I recommend publication as is.

Reviewer #3 (Remarks to the Author):

My questions have been answered.

Reviewers' comments:

Reviewer #1 (Remarks to the Author):

The authors have prepared additional explanations and have revised their manuscript, according to my queries and requests. However, I have to report that it is difficult for me to accept the authors' answers and revisions because of the following reasons;

1) On the interpretation of ADF-STEM image contrasts related to defects in graphite;

The authors state difficulties of image simulation for understanding the STEM image contrasts because of many experimental uncertainties. However, if it is not proved that the significant contrasts, i.e. paired circles and bright/dark bands, are coming from defects at least qualitatively, it is impossible to discuss about the relationship between the intercalation events and the behaviors of defects the authors have claimed. Based on such a speculation that the characteristic STEM contrasts are caused by the particular defects, the discussions and conclusions here should be also considered to be nothing more than speculations. I believe that a clear evidence is necessary on this point because the role of the defects is one of the most important issues in this article.

We are happy to learn that we answered many of your questions with our first response. We hope that we can address your remaining concerns here.

Without more detailed knowledge of the structure of the sample, we cannot perform a well-constrained simulation of our images. However, we do have a qualitative understanding of our image contrast, and we can demonstrate that our STEM imaging shows defects in the graphite.

We use STEM imaging conditions that might appear unusual to some microscopists: because we are not attempting to achieve high resolution, we can work with relatively small convergence angles, which more nearly approximate the parallel beam condition of TEM. We chose to work with such convergence angles in the hopes that we could maximize the contribution to our dark field signal from the graphite's first order Bragg peaks, which are especially sensitive to the graphite stacking (see DOI: 10.1103/PhysRevB.87.045417). Graphite's stacking is thought to sometimes shift from AB in pristine material to AA when fully intercalated (DOI: 10.1080/00018730110113644, p. 36), and we wanted to maximize our sensitivity to such a shift.

While in the end we did not observe the AB to AA shift, our choice of convergence angle and camera length makes our ADF STEM images much more like TEM images than is usual. We have added a new figure to the supplementary information that shows, alongside STEM images that already appear in the manuscript, TEM images acquired under two different dark field (DF) conditions (i.e. using an objective aperture to select graphite's 1st and 2nd order Bragg peaks respectively). While the contrast is different in all three cases, features appearing in the ADF STEM images have their clear counterparts in the DF TEM images. In particular, the circles and bright/dark bands in the ADF STEM images are associated with defects (i.e. regions of non-uniform contrast as would be generated by deviations from perfect crystallinity) in the TEM images. The one-to-one correspondence between the defects observed in TEM and the particular features seen with ADF STEM demonstrates that these features are, in fact, crystal defects as claimed.

The changes made to the manuscript to clarify these points are enumerated in our response to (3) below.

2) On the effects of electron beam irradiation;

I agree with the authors that the intercalation events are fundamental because the structural changes and the correlated current spikes are detected in both STEM and OM observations. However, this does not lead to that e-beam does not affect to the reaction. The authors mentioned that beam-induced phenomena can be distinguished from intercalation-induced ones because beam-induced damages (e.g. changes to amorphous) arise gradually. I cannot neglect possibilities that abrupt events happen due to the e-beam, for example, focused electron probe may release pinning sites between the graphite sheets, which is not periodic phenomenon. Therefore, for evaluating the effect of e-beam, it is important to compare images taken before/after the intercalation reaction at regions where e-beam is irradiated / not irradiated during reaction.

You are correct: we are not able to rule out the possibility that the electron beam, for example, releases pinning sites and thus plays a role in the determining the timing of the abrupt events observed. However, the existence of such a beam-sample interaction would strengthen our main conclusion, which is that the defects are dominating the transport at the single crystal level. For in this case, the energy thresholds for unpinning would be even higher without the beam, and the transport would feature even more marked abrupt events. (We note, however, that we looked for and did not see more marked abrupt events without the beam, as will be discussed below.)

Specific data of the type requested, namely successive images where the imaged region was and was not irradiated during intercalation are, in fact, presented in the paper. Figure 5 shows part of the sample of Movie 3 above the fold before cycling, and after 1, 3, 4, 8, and 9 complete intercalation cycles. During cycles 5, 6, 7, and 8 this flake was being imaged in a different region. Other than some slight additional debris in the field of view, the image (Fig. 5e) acquired after the 8th cycle (i.e. at the beginning of the 9th cycle – this language has been clarified in this revision) does not seem special in any way. The defect structure seen is somewhat changed from what was seen earlier and later, but the same can be said of any of these images. Given that we cannot image the sample without exposing it to the electron beam, it is not clear to us how to identify a signature of beam-sample interactions in the STEM image data.

We do have strong evidence, however, that the beam is not playing a dominant role in the reaction chemistry. The current-voltage data with the beam blanked (Supplementary Figure 8 for this sample) has no characteristics that distinguish it from those acquired as the sample is being imaged with the beam (Supplementary Figure 7 for this sample). As the CV data acquired without the electron beam shows the abrupt events, it demonstrates that the beam is not responsible for the abrupt events. While it does not rule out beam-induced chemistry entirely, this control experiment indicates that the role of such chemistry in the overall process must be small.

To clarify our reasoning in the manuscript, we have added the following text to the section

Defect re-organization over multiple cycles: “While we cannot rule out the possibility that the electron beam precipitates abrupt events (for example, the focused probe may release defects pinning graphite layers), we note that in such a case the charge transfer would be dominated by defects even in the absence of the triggering electron beam, with larger thresholds. (Such a difference in the transport between the beam-on and beam-off conditions was, however, not observed – see Supplementary Figure 8.)” **We have also added the following text to the caption for Supplementary Figure 8:** “These curves are very similar to those generated while the sample was being imaged (shown in the previous two supplementary figures), which demonstrates that the imaging electron beam played, at most, a minor role in determining the electrochemical transport.”

3) On the domain structure appeared as bright/dark narrow bands;

The authors explained that these contrasts in Fig.6 showing large aspect ratio may be produced by imperfect illumination tilted from normal incidence. However, this explanation is not valid because it’s for the bright-field TEM imaging. The authors should discuss the contrast in the ADF-STEM imaging mode they have utilized.

As explained in our response to point (1), our STEM beam had a small convergence angle which gives images with TEM-like contrast. In the Methods section we have expanded what was one sentence into two to explain this point,

replacing “The camera length was adjusted to eliminate signal from the central beam while maximizing signal from the first order diffraction peaks of the single crystal graphite.”

with “The camera length and convergence angle (2.9 mrad) were chosen to maximize the ratio of the ADF detector signal from the first order relative to the second order graphite Bragg peaks, while excluding the zeroth order peak entirely. Although the small convergence angle precludes high-resolution imaging, these imaging conditions give both enhanced sensitivity to AB-to-AA stacking changes³¹ (thought to sometimes occur in the stage 1 to stage 2 transition¹) and TEM-like contrast (see Supplementary Figure 9), which were more valuable for this study.”

We have also added the new Supplementary Figure just mentioned. This figure shows two different regions of the graphite flake of Figure 4, Figure 5, Supplementary Figures 1a and 5a, and Supplementary Movie 3, each imaged in three different modes: ADF STEM, 1st order DF TEM, and 2nd order DF TEM. Comparing these six images side-by-side shows that the STEM contrast is similar to the TEM contrast (note, for instance, that the moiré pattern from the fold gives lines, i.e. high-aspect ratio features, in both STEM and 2nd order DF TEM), and that features (e.g. defects) seen in DF TEM also appear in STEM.

Reviewer #2 (Remarks to the Author):

The authors have addressed my previous concerns; thus I recommend publication as is.

Reviewer #3 (Remarks to the Author):

My questions have been answered.

We are gratified to see that the questions and concerns of both Reviewer #2 and Reviewer #3 have been satisfactorily resolved.

Reviewers' comments (final round of peer review)

Reviewer #1 (Remarks to the Author):

The authors added some helpful data and detail explanations to answer my requests. However, I think the authors still misunderstand the image contrast and the interpretation, and the image quality is very poor to argue the conclusion. The detailed reason is shown as follows:

The authors stated that the STEM imaging condition they have purposely adopted, i.e. small convergent angle, is approximate to illumination condition in TEM (approximately parallel incidence). In fact, related contrast may appear when compares the ADF-STEM image and the 1st order DF-TEM image, as shown in Supplemental Figure 9. Such correspondences are, however, incorrect from the viewpoint of the electron optics. According to the reciprocal theorem between TEM and STEM images, decrease of the probe-convergent angle in STEM corresponds to decrease of the collection angle by reducing the size of objective aperture in TEM. On the other hand, the images obtained with the (semi-)parallel illumination (decrease of electron-source size) in TEM correspond to those with the (semi-)infinite small bright-field detector (decrease of detection angle) in STEM. If the authors here would like to achieve the TEM-like imaging conditions in STEM, a tiny bright-field detector should be adopted.

Furthermore, the authors have claimed that both DF-TEM and the present ADF-STEM imaging conditions are sensitive to graphitic AA and AB stacking structures. However, image contrast in the above imaging is rather sensitive to local Bragg conditions. The constant periodicity of the bright/dark bands is hardly obtained in any (S)TEM images of graphite samples due to small bends in the graphitic layers, especially for the partially intercalated ones. I'm afraid that any quantitative interpretations must be very difficult because many factors affect the image contrast in the present ADF-STEM condition arising from the rather large ADF detector (even small-angle diffracted beams are interfered and detected). It is therefore recommended to utilize HAADF detectors for more simplicity.

Reviewers' comments (final round of peer review)

Reviewer #1 (Remarks to the Author):

Our point-by-point response to Referee #1's most recent comments is given here:

The authors added some helpful data and detail explanations to answer my requests. However, I think the authors still misunderstand the image contrast and the interpretation, and the image quality is very poor to argue the conclusion.

The detailed reason is shown as follows: The referee is gracious to acknowledge that our additions have been helpful. We understand the image contrast well enough to support our interpretations. An unbiased observer might criticize the image quality in movies SM2 and SM3 as only fair (though we note that the SM3 sample was imaged for an unprecedented 9 complete intercalation cycles over more than 3 hours). The image quality in the others (SM4 and SM5) is excellent. We present no data that can fairly be described as "very poor".

The authors stated that the STEM imaging condition they have purposely adopted, i.e. small convergent angle, is approximate to illumination condition in TEM (approximately parallel incidence). In fact, related contrast may appear when compares the ADF-STEM image and the 1st order DF-TEM image, as shown in Supplemental Figure 9.

These statements are all true, and constitute agreement with the point we were trying to make in our previous response: our STEM imaging conditions give contrast that is related to DF-TEM contrast. Thus the defects that we see in STEM are the same defects that are revealed by TEM.

Such correspondences are, however, incorrect from the viewpoint of the electron optics. According to the reciprocal theorem between TEM and STEM images, decrease of the probe-convergent angle in STEM corresponds to decrease of the collection angle by reducing the size of objective aperture in TEM. On the other hand, the images obtained with the (semi-)parallel illumination (decrease of electron-source size) in TEM correspond to those with the (semi-)infinite small bright-field detector (decrease of detection angle) in STEM. If the authors here would like to achieve the TEM-like imaging conditions in STEM, a tiny bright-field detector should be adopted.

Our goal was not to achieve STEM contrast identical to what is seen in TEM, or more specifically, bright field TEM (BF TEM). Our goal was to achieve sensitivity to structural changes. We selected a convergence angle and camera length such that the first order diffraction peaks struck our annular dark field (ADF) STEM detector, and thereby achieved contrast similar to that seen with dark field TEM (DF TEM). We used an annular dark field (ADF) STEM detector, which has as its TEM conjugate (using the reciprocity theorem) the less-common annular ("hollow cone") illumination aperture to provide DF TEM analogous imaging conditions (see "Hollow cone illumination for fast TEM, and outrunning damage with electrons" by Spence, Subramanian, and Musumeci, doi:10.1088/0953-4075/48/21/214003). The referee's description of the contents of the reciprocity theorem is perfectly correct, but not applicable to our imaging conditions. The referee describes a BF TEM setup, but we restricted ourselves to DF TEM

specifically to single out diffraction from graphite and reject other signals. We are saying that, using this DF TEM analogous imaging technique, we can see defects with STEM, which the referee does not dispute. And we are saying that our STEM contrast is TEM-like, which we support by showing TEM images side-by-side with STEM images (Supplementary Figure 9).

Furthermore, the authors have claimed that both DF-TEM and the present ADF-STEM imaging conditions are sensitive to graphitic AA and AB stacking structures. However, image contrast in the above imaging is rather sensitive to local Bragg conditions. The constant periodicity of the bright/dark bands is hardly obtained in any (S)TEM images of graphite samples due to small bends in the graphitic layers, especially for the partially intercalated ones. I'm afraid that any quantitative interpretations must be very difficult because many factors affect the image contrast in the present ADF-STEM condition arising from the rather large ADF detector (even small-angle diffracted beams are interfered and detected). It is therefore recommended to utilize HAADF detectors for more simplicity.

Again, these statements are unconnected to any of the conclusions we draw in the manuscript. We mention AA and AB stacking precisely once in the manuscript, and then only in the Methods section. We do not otherwise discuss stacking anywhere. None of our conclusions have anything to do with stacking. As Referee #1 notes, quantitative interpretation of our contrast is difficult. Since quantitative interpretation was not our goal, we chose imaging conditions that we deemed likely to be most sensitive to changes in the graphite structure. Our choice was successful: in a series of movies we see defects move, disappear, and rearrange repeatedly over numerous cycles. Our conclusions are based on these simple, well-supported, and now uncontested observations. HAADF imaging, which we tried, gives smaller signals with a poorer signal-to-noise ratio. This result is unsurprising, since HAADF imaging is ill-suited for low-Z materials such as graphite and sulfuric acid. Furthermore, while HAADF is sensitive to gross changes in the amount of material present, for a typical intercalation event this change is small. Thus HAADF imaging cannot effectively reveal the structural changes that were our primary interest.

Reviewers' comments:

Reviewer #4 (Remarks to the Author):

I agree with the authors that the key point of the paper is that intercalation events are stochastic in nature, and this is a very important result that changes our thinking about this process. The comments from Referee #1 indicate he/she is confused about the nature of the contrast. The authors referred to the contrast as TEM-like. It would be better to just call the contrast diffraction contrast, which is maximized by the small probe angle (hence no atomic resolution) and their choice of detector angle to include first order diffracted beams. The referee does not seem to realize that ADF imaging is sensitive to diffraction contrast, hence the confusion. It is definitely not required to understand the detailed nature of the contrast to justify the main, important, claim of the paper. Clearly, if a lattice is being deformed by intercalating species, there will be some change in the relative positions of the graphite planes, and diffraction contrast would be sensitive to this. The paper should be published and further modeling of the nature of the lattice deformation can be studied later.

I would like to make a couple of suggestions. The phrase "crystal-spanning" is unfamiliar to me, perhaps the authors could define it. Also, their claim that "STEM's serial pixel acquisition improves on the time resolution of a full-frame acquisition by almost three orders of magnitude" seems excessive. Are there not fast cameras available these days capable of thousands of frames per second?

Reviewers' comments:

Reviewer #4 (Remarks to the Author):

I agree with the authors that the key point of the paper is that intercalation events are stochastic in nature, and this is a very important result that changes our thinking about this process. The comments from Referee #1 indicate he/she is confused about the nature of the contrast. The authors referred to the contrast as TEM-like. It would be better to just call the contrast diffraction contrast, which is maximized by the small probe angle (hence no atomic resolution) and their choice of detector angle to include first order diffracted beams. The referee does not seem to realize that ADF imaging is sensitive to diffraction contrast, hence the confusion. It is definitely not required to understand the detailed nature of the contrast to justify the main, important, claim of the paper. Clearly, if a lattice is being deformed by intercalating species, there will be some change in the relative positions of the graphite planes, and diffraction contrast would be sensitive to this. The paper should be published and further modeling of the nature of the lattice deformation can be studied later.

We thank the reviewer for acknowledging the importance of this work. “TEM-like” has been replaced with “diffraction” per this suggestion.

I would like to make a couple of suggestions. The phrase "crystal-spanning" is unfamiliar to me, perhaps the authors could define it.

We have re-written the section where “crystal-spanning” is introduced to make this concept clearer.

Also, their claim that "STEM's serial pixel acquisition improves on the time resolution of a full-frame acquisition by almost three orders of magnitude" seems excessive. Are there not fast cameras available these days capable of thousands of frames per second?

The referee is correct; there are fast cameras for TEM available today. But these cameras, when used for STEM, are still slow in comparison to a STEM detector. Our intent here was only to compare the line-to-line temporal resolution to the frame-to-frame resolution within our (or really any) STEM imaging system. However, we feel adding additional clarification here would distract from the main message of the paragraph, so we have elected to remove this sentence from the manuscript.